# communications
# engineering

# Celestial compass sensor mimics the insect eye for navigation under cloudy and occluded skies

Evripidis Gkanias [1]✉, Robert Mitchell [1], Jan Stankiewicz[1], Sadeque Reza Khan [2], Srinjoy Mitra [3] & Barbara Webb [1]

Insects use the sun's position (even when concealed) as a compass for navigation by filtering celestial light intensity and polarisation through their compound eyes. To replicate this functionality, we present a sensor that imitates essential aspects of insect eyes, particularly the fan-like arrangement of polarised light receptors in their dorsal rim area. Our sensor comprises a ring of eight pairs of photodiodes (evaluating two orthogonal orientations of polarised light) to analyse the skylight coming from different directions. Because the layout of our sensor aligns with the polarised light pattern in the sky, a circular-mean model that integrates information spatially across the analysers can estimate the solar azimuth. When using the same sensor design, our model achieves lower compass errors than alternative (and computationally more complex) algorithms, especially under cloudy and occluded skies. Thus, the morphology and processing of the insect celestial compass provide an efficient and robust directional input for navigation.

[1] School of Informatics, University of Edinburgh, Edinburgh, United Kingdom. [2] School of Engineering and Physical Sciences, Heriot-Watt University, Edinburgh, United Kingdom. [3] School of Engineering, University of Edinburgh, Edinburgh, United Kingdom. ✉email: ev.gkanias@gmail.com

A compass is used in navigation to provide robust orientation estimates. As travel distance increases, idiothetic orientation information—from proprioception or an inertial measurement unit (IMU)—will eventually succumb to noise[1]. Typical allothetic compass solutions such as magnetometers often suffer from electronic interference, and the alternative of relying on a global positioning system (GPS) creates dependence on a full infrastructure of satellites which may not always be available. Ideally, a compass should be independent of external support, impervious to disturbance, lightweight, cheap, and energy efficient. The celestial compass (inspired by insects[2–4]) fits all these criteria.

A celestial compass uses the properties of light coming from the sky to estimate the position of the primary celestial body (herein, the sun) or, with appropriate time compensation, the North. The light from large celestial objects (e.g., the sun or moon[5]) creates a regular pattern of intensity and polarisation across the sky (Fig. 1a). Under clear skies, skylight intensity peaks at the visual position of the sun (solar azimuth and elevation), while the degree of polarisation (DoP) is strongest in the opposite direction (at 90° from the sun and across the zenith). At a specific point, the angle of polarisation (AoP) is always perpendicular to the arc between the sun and that point, forming concentric rings of polarisation around the visual position of the sun[6–8]. These stereotyped skylight patterns allow for various methods to recover the sun's position.

A range of celestial compass sensors has been developed that integrate AoP estimations from different positions in the sky to locate the sun (using photodiodes[9–13], or cameras[14–21]). Theoretically, one can locate the sun by measuring the AoP in two points of the concentric-rings pattern. These sensors focus on the accurate extraction of the AoP and ignore the remaining (useful) properties of light. Estimating the AoP in silico is straightforward[22] (using at least three polarisation filters and photodiodes), but the complication arises when attempting to compute the solar azimuth, which requires computing eigenvectors[9–21], a discrete Fourier transform (DFT)[23–25], or finding the straightest line of AoPs passing through the zenith[26,27].

The eyes and brains of many insects evolved to provide an alternative solution[28,29]. Each of their compound eyes has near 180° panoramic vision (desert ants[30], fruit flies[31]) and is formed by hundreds of facets called ommatidia (Fig. 1b). The ommatidia that occupy the dorsal rim area (DRA) of the eyes are sensitive to polarisation. Each dorsal rim ommatidium analyses the incident light in two polarisation axes: one roughly perpendicular to its meridian from the dorsal-most position on the eye (orange in Fig. 1b–d; $I_{90}$), and one parallel to this meridian (blue in Fig. 1c, d; $I_0$). The fan-like arrangement of DRA ommatidia[3] (see Fig. 1b, e) forms what Wehner called a matched filter[32] that imitates the sky AoP spatial pattern and partially solves the navigational problem by transferring the complexity from algebra to geometry.

Lambrinos et al.[33,34] first developed a sensor that could perceive and process light in a similar way to the DRA ommatidia (Fig. 1f). They replicated an ommatidium by using two photodiodes behind orthogonal polarisation filters (similar to Fig. 1d) and integrating their signals to the response of polarisation-opponent (POL-OP) units (Fig. 1g). The response of three units (pointing at the sky zenith but oriented at 120° intervals) was used to estimate the AoP at the zenith, predicting two possible solar azimuths that are 180° apart. They resolved this ambiguity by comparing the intensity of light in these two directions from surrounding polarisation-insensitive photodiodes. A similar bio-inspired approach has since been followed by others[22,24,25,35–40]. However, Smith[41] observed that rotating a (simulated) POL-OP unit at an angle from the zenith breaks the 180° ambiguity and its

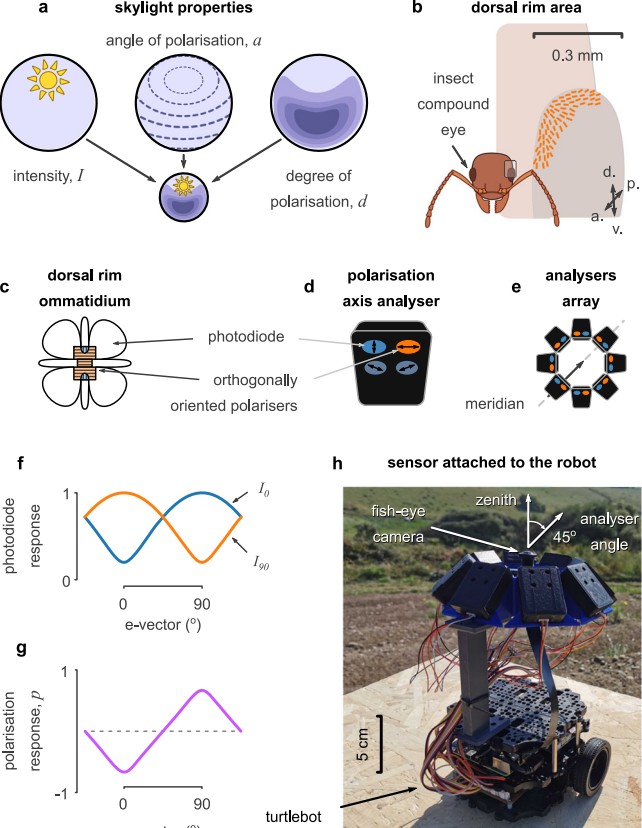

**Fig. 1 Overview. a** Skylight varies in light intensity (*I*), angle (*a*) and degree (*d*) of polarisation. **b** The compound eyes of insects filter light using specialised ommatidia in the dorsal rim area (DRA). These form a fan-like distribution, pointing in different directions in the sky, and covering up to 120° field of view. d.: dorsal, v.: ventral, p.: posterior, a.: anterior. **c** In each ommatidium, the skylight is filtered by orthogonally oriented microvilli before it is captured by two groups of photoreceptor cells. **d** Our polarisation axis analyser (PAA) mimics this function by filtering the skylight with two orthogonal polarizers before being captured by the photodiodes. **e** Our sensor has eight PAAs in a ring arrangement, elevated by 45°. **f** The responses of the orthogonal photodiodes ($I_0$ and $I_{90}$) over the e-vector of polarised light. **g** The normalised difference between the orthogonal responses (*p*) matches the response of the polarisation-sensitive neuron in the insect optic lobes. **h** Our robot uses a panoramic camera and our designed sensor to collect skylight data. **c**, **f**, and **g** were adapted and modified from[29].

response is strongest in the opposite direction of the solar azimuth (corresponding with the DoP pattern in the sky). Therefore, they suggested that the fan-like arrangement of the dorsal rim ommatidia could have evolved to take parallel measurements of the sky polarisation at different positions, obtaining the information from a single reading (without scanning) and inferring an instantaneous and unambiguous estimate of the sun's direction. Gkanias et al.[29] independently arrived at the same idea and provided proof of principle for such a celestial compass using a simulated sensor array that mimicked the layout of the insect DRA.

Here, we extend this line of work and provide a hardware prototype of the celestial compass sensor of insects. This sensor may be used to determine relative solar azimuth using either polarised light or intensity information. We validated the performance of our compass model by recording sensor data under a wide range of atmospheric, weather, and sky occlusion

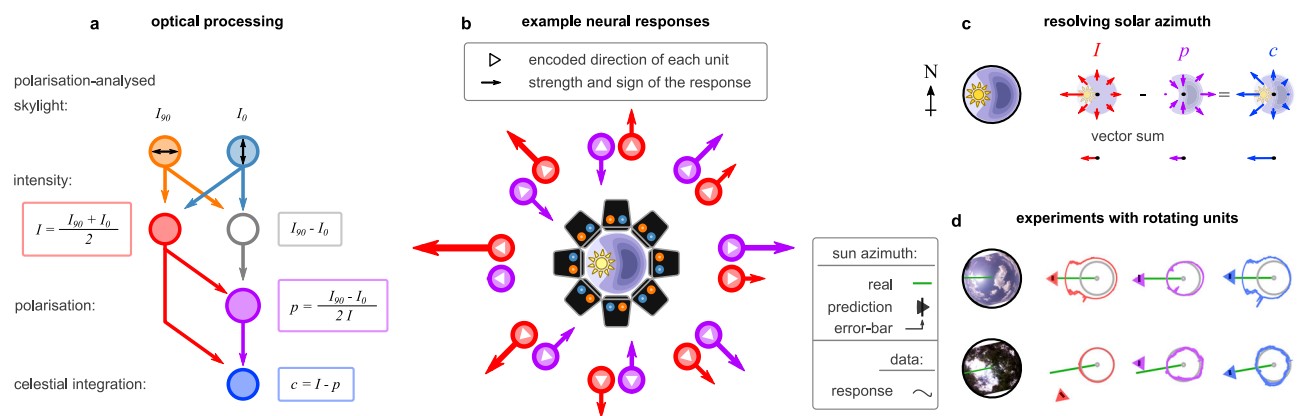

**Fig. 2 Description of the computational model. a** Adding the pulses from two photodiodes (PDs) with orthogonal polarised light filters can approximate the light intensity ($I$); subtracting them approximates the polarisation Fresnel ratio (PFR). Normalising the PFR with light intensity provides a good approximation of how well the polarity of light is aligned with the orientation of the polarisation axis analyser (PAA, $p$). The difference between $I$ and $p$ results in their celestial integration ($c$), which indicates how well the PAA is aligned with the brightest celestial body in the sky. **b** An example of the intensity and polarisation measurements of all the PAAs in a clear sky condition. Each measurement can be multiplied with a unit vector directed towards the azimuth of the PAA (see red and purple arrows), which can also flip by negative measurements (e.g., for $p$). **c** The simple addition of all the vectors of each unit type results in a vector that has the direction of the solar azimuth (anti-solar for $p$). **d** The measured responses of the different types of units (solid lines) in two different sky conditions (clear and partially cloudy with occlusions from trees) and the estimations of our model using only $I$ (red), $p$ (purple), or $c$ (blue). Small triangles denote the median estimation across a set of 360 homogeneously distributed facing directions (one per degree) of the sensor; circular lines denote the first and third quartiles of the distribution; the green line highlights the solar azimuth.

conditions. We also compared its performance with several alternative models that could use similar input. Our results confirm that the proposed sensor provides a robust compass system that could be used by autonomous vehicles, and also provides insight into the neural processing stages underlying the celestial compass of insects.

## Results

**The celestial compass sensor**. The sensor consisted of an array of eight polarisation axis analysers (PAAs) distributed evenly around a ring, then tilted to give an inclination of 45° (Fig. 1e, h). This arrangement (loosely) mimics the fan-like arrangement of the dorsal rim ommatidia across both insect eyes. Each PAA measured the voltage from four ultraviolet (UV) sensitive photodiodes (Fig. 1d), which were covered by linear polarisers in a different orientation (0°, 45°, 90°, and 135°). Readouts from all four photodiodes were recorded, but to maintain equivalence to the insect eye, our model used only two—those sensitive to light polarised at 0° and 90°, which correspond to $I_0 \in [0, 1]$ and $I_{90} \in [0, 1]$ respectively (Fig. 2a).

Following Stürzl[16], the overall light intensity is

$$I = (I_{90} + I_0)/2. \tag{1}$$

Polarisation information is calculated as the intensity-normalised POL-OP[29], which ensures that $p \in [-1, 1]$,

$$p = (I_{90} - I_0)/(2I). \tag{2}$$

This value will be maximal when the PAA is perpendicularly aligned with the AoP of the light source (negative when parallelly aligned) and is proportional to the DoP (see Fig. 1g). Subtracting $p$ from $I$ gives what we call the celestial integration,

$$c = I - p. \tag{3}$$

Under clear sky conditions, $I$ is higher when the PAA points towards the sun (solar azimuth) and is almost uniform across the remaining directions (Fig. 2b). In contrast, $p$ is highest at the anti-solar azimuth (where the DoP is highest and the AoP is aligned with the filter of $I_{90}$), and negative at the sides and closer to the solar azimuth. A local maximum in $p$ occurs at the solar azimuth

but is usually lower than the one in the anti-solar azimuth. Thus, the celestial integration ($c$) is higher towards and around the solar azimuth and gradually decreases towards the anti-solar azimuth (Fig. 2c). Although $c$ looks similar to $I$ under clear sky conditions, it becomes more informative than $I$ about the solar azimuth under cloudy or occluded skies (Fig. 2d).

To generate a compass signal from the array, the values from each PAA were integrated according to the model suggested by Gkanias et al.[29]. Values ($c$) were transformed into vectors with their polar angles equal to the azimuth of their recording PAAs and lengths equal to the values themselves. The vectors were then averaged, yielding a mean vector that points towards the solar azimuth. If the celestial integration of the $k^{th}$ PAA is $c_k$, the mean vector can be calculated as a complex number,

$$z_c = \frac{1}{K} \sum_{k=1}^{K} c_k \, e^{i 2\pi(k-1)/K}, \tag{4}$$

where $2\pi(k-1)/K$ represents the azimuth of the $k^{th}$ PAA for $K$ PAAs evenly distributed around the ring. The response coming from each PAA is interpreted as its estimation of whether the sun is in its facing direction. Negative responses would indicate that the sun is estimated to be in the opposite direction (see Fig. 2b, c, d). Note that the mean vector also provides a measure of dispersion in its magnitude[42,43]. The mean solar azimuth indicated by each PAA ($\alpha_c$) and the angular deviation of these values ($\sigma_c$) can therefore be recovered as

$$\alpha_c = -i \ln \frac{z_c}{||z_c||}, \qquad \sigma_c = \sqrt{2(1 - ||z_c||)}, \tag{5}$$

where $||z_c||$ is the length of the mean vector. By replacing $c_k$ with $I_k$ or $-p_k$ in (4), we may estimate the solar azimuth based only on intensity ($z_I$) or polarisation ($z_p$) respectively. Depending on the values used we could have an intensity, polarisation, or celestial compass.

Figure 2d shows examples of $I$, $p$, and $c$ values, along with their respective estimates of the solar azimuth for two different sky conditions. Under an almost clear sky, all provided similar estimates. However, with canopy occlusion, the intensity and polarisation estimates show a consistent deviation towards

opposite sides of the foliage opening, which was corrected by the celestial integration.

**A dataset of varying skylight conditions**. Our sensor was mounted on a robot to collect data from remote sites (Fig. 1h and Materials and Methods). Data were collected from field sites in Italy and South Africa (in May and November 2022 respectively), and our dataset comprises a variety of solar elevations, weather conditions, and occlusions. The collection was organised into sessions, each of which consisted of twelve complete (360°) rotations under a given condition. At the beginning of each recording session, the robot was initialised facing magnetic North (±5°) and IMU was reset to zero. On each rotation, the robot would log all photodiode responses and IMU data (to measure the deviation from the starting direction; Supplementary Fig. 1a). We computed the angle of the facing direction of the robot with respect to the solar azimuth by taking the difference between the IMU and the theoretical solar azimuth (for the particular location, time and day of the recording). This raw dataset was then transformed into a pooled dataset which additionally included the responses of each optical unit ($I$, $p$, and $c$; Supplementary Fig. 1b).

Using this pooled dataset we were able to reconstruct the performance of the same basic design using different numbers of PAAs distributed evenly around the sensor ring (Supplementary Fig. 1c). This was achieved by determining the preferred direction of the reconstructed PAAs, then taking the median of the five responses from the dataset which were most closely clustered around that preferred direction (Supplementary Fig. 1d).

**Spatial sampling**. Using the interpolation process described in Supplementary Fig. 1, we compared the performance of our sensor in predicting the solar azimuth, for varying numbers of (hypothetical) PAAs. We refer to solar azimuth prediction error as the global reference error as it represents how well we could predict the actual solar azimuth from one compass sensor reading (from the given number of PAAs). However, this measure is subject to the ±5° error incurred when initialising the robot (note that attempting to use the fish-eye images to determine the actual solar azimuth was subject to even more error, due to lens distortion, lens flare and occlusions). We therefore also evaluate the local reference error which is how well the sensor can estimate the robot's angular displacement, for any point in its rotation, relative to its starting direction, where the IMU measurement of displacement (which has negligible error over the short time scale) is taken as ground truth.

To assess the effect of PAA number, we selected the easiest scenarios from our dataset (little to no cloud, solar elevation at least 10°); the results are shown in Fig. 3a, b and Supplementary Table 1. With the minimum of three PAAs, the root mean square error (RMSE) for the sun direction (global reference) was relatively high (10.53°). The lowest error was achieved with our maximum of sixty PAAs (RMSE 2.65°), i.e., within the range of the initialisation error. Adding PAAs always improved the performance. The local reference error was high for fewer than six PAAs but dropped substantially for six or more (3.78°, see Supplementary Table 1). Beyond thirty-five PAAs, performance improvements were negligible (RMSE 0.53° for 36 PAAs, compared to 0.43° for 60 PAAs). In all subsequent results, we use only eight PAAs as in our hardware implementation.

**Solar elevations**. Under (relatively) clear sky conditions, we examined the effect of solar elevation on sensor performance. Data were sorted by their solar elevation and binned in 5° intervals from -5° (below the horizon) to 85° (near zenith). Figure 4a shows examples of the sensor performance for specific

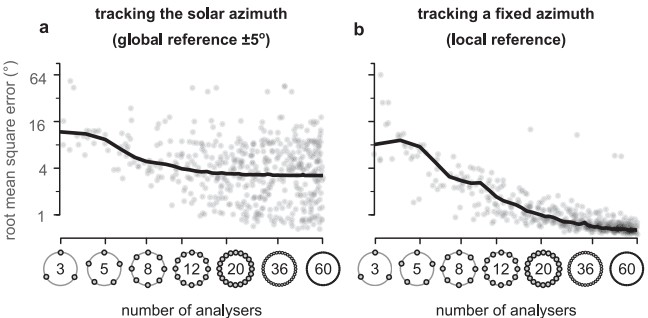

**Fig. 3 Sensor performance over variable spatial sampling. a** Root mean squared error (RMSE) of estimating the solar azimuth (±5° error). **b** RMSE of tracking any fixed direction. Vertical axes are in the $\log_2$ scale. Horizontal axes are in $\log_6$ scale, and random noise (0.5) was added to the (whole) number of polarisation axis analysers, making them appear as a continuous distribution. Ticks without labels denote (from bottom to top) 0°, 2°, 8°, 32°, and 90° RMSE. Although the spread appears to be increasing with higher spatial resolution, this is an illusion of the logarithmic scale of the vertical axis. Standard deviation (SD) in **a** is 21.77° for 3 analysers and it falls in the range from 4.41° to 5.15° from 4 to 60 analysers (randomly distributed). The SD in **b** is 21.92° for 3 analysers and it falls in the range from 2.82° (4 analysers) to 0.48° (60 analysers; decreasing for higher spatial resolutions) for the rest of the cases.

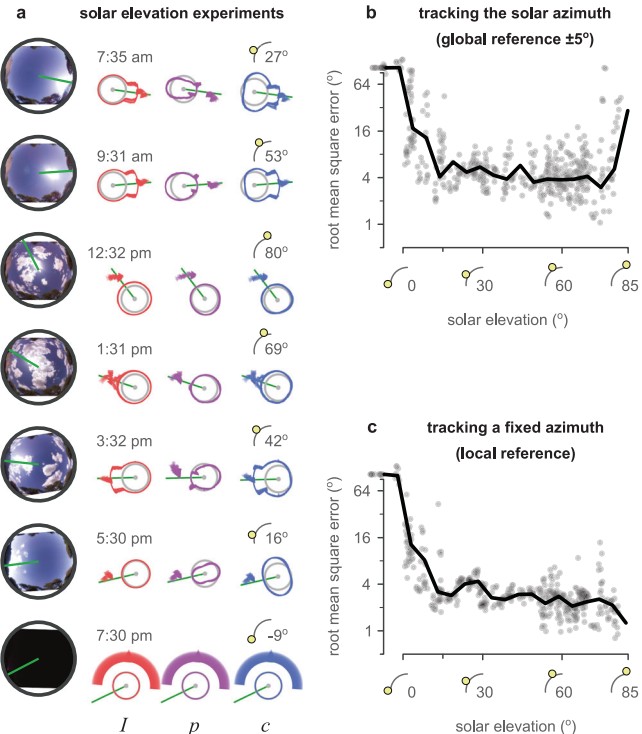

**Fig. 4 Sensor performance over different solar elevations. a** Examples of different solar elevations in the sky. Left: fish-eye images of the sky for each example. Right: $l$, $p$, and $c$ values in each example, the prediction of the model using these responses (intensity, polarisation, and celestial compass, respectively). Above the responses are the time of day and the solar elevation during the session. Each example shows the median (arrowhead) and quantiles (circle segment) of the 360 predictions produced (at homogeneously distributed orientations) during each of the twelve full rotations of the sensor. **b** Root mean squared error (RMSE) of estimating the solar azimuth (±5° error). **c** RMSE of tracking any fixed direction. Vertical axes are in the $\log_2$ scale. Ticks without labels denote (from bottom to top) 0°, 2°, 8°, 32°, and 90° RMSE.

solar elevations. Figures 4b and c show respectively the global and local reference errors for all the sessions, which are also summarised in Supplementary Table 2. There was no noticeable difference in the performance of our sensor for solar elevations above 10° (average RMSE, global: 5.89°, local: 2.77°). Lower elevations caused the error to increase rapidly; at and below 0° elevations the sensor was unable to provide useful predictions. There was also a sharp increase in the global error at high elevations that was not reflected in the local error; we believe this might reflect the increasing inaccuracy of our (hand-selected) estimate of the true sun direction in the sky images as the sun nears the zenith, rather than inaccuracy in the compass per se. Overall, the sensor performance was stable throughout the day when the sun was higher than 10° from the horizon.

**Atmospheric conditions**. As well as different sky conditions, our data were collected in three locations that represent different atmospheric conditions: Sardinia (Italy), Vryburg (South Africa), and Bela Bela (South Africa)—see Materials and Methods for more details. Sardinia is an island and the data were collected close to the shore, where the humidity was generally high. Vryburg is located in the African Savannah, where the climate is dry, and Bela Bela is in the woodlands, which provides an average climate. Supplementary Table 3 shows that the performance of our compass sensor and model was not affected by the climates of these different locations. Interestingly, the panoramic images of the sky that we collected from Sardinia look surprisingly similar to the ones reported in oceanic atmospheric conditions[44]. Thus, they could approximate off-shore atmospheric conditions and suggest that our sensor could be used in intercontinental missions.

**Cloudy skies**. The examples in Fig. 4a suggested that (under clear skies) the predictions of solar azimuth made by our compass model were unaffected by the value used (I, p, or c). Under a clear sky, the relationship between solar azimuth, light intensity, and polarisation remains stable. However, the presence of clouds disrupts this relationship. The thickness of clouds determines how sunlight is scattered, and affects both the polarisation and intensity distribution in the sky. Our dataset includes recording sessions under a variety of sky conditions which we classified based on their cloud cover (for examples see Fig. 5a). Global and local error under varying cloud conditions can be seen in Fig. 5b, c and Supplementary Table 4. Sensor function was not substantially disrupted by thin or broken cloud cover but deteriorated for thicker cloud cover (uniform and solid).

**Occluded skies**. Some insects which are known to use polarisation for orientation live in densely wooded areas, so an insect-inspired celestial compass should function under canopy cover. The same principle extends to robots working in built-up areas; a celestial compass should be robust to local occlusion due to buildings. We tested our sensor with different degrees of canopy occlusion (examples can be seen in Fig. 6a). Results are shown in Fig. 6b, c and Supplementary Table 5. In general, the intensity and polarisation compasses showed a strong dependence on whether the solar or anti-solar side of the sky was blocked respectively. The celestial integration could effectively use information from whichever was most useful at the time. Where trees occupied the central area or full panorama (with relatively open foliage), polarisation tended to be more robust than the intensity or celestial compasses. Additional PAAs generally reduced both global and local RMSE (especially for the intensity compass, and for scenarios close to trees and with dense cover; see Supplementary Table 5).

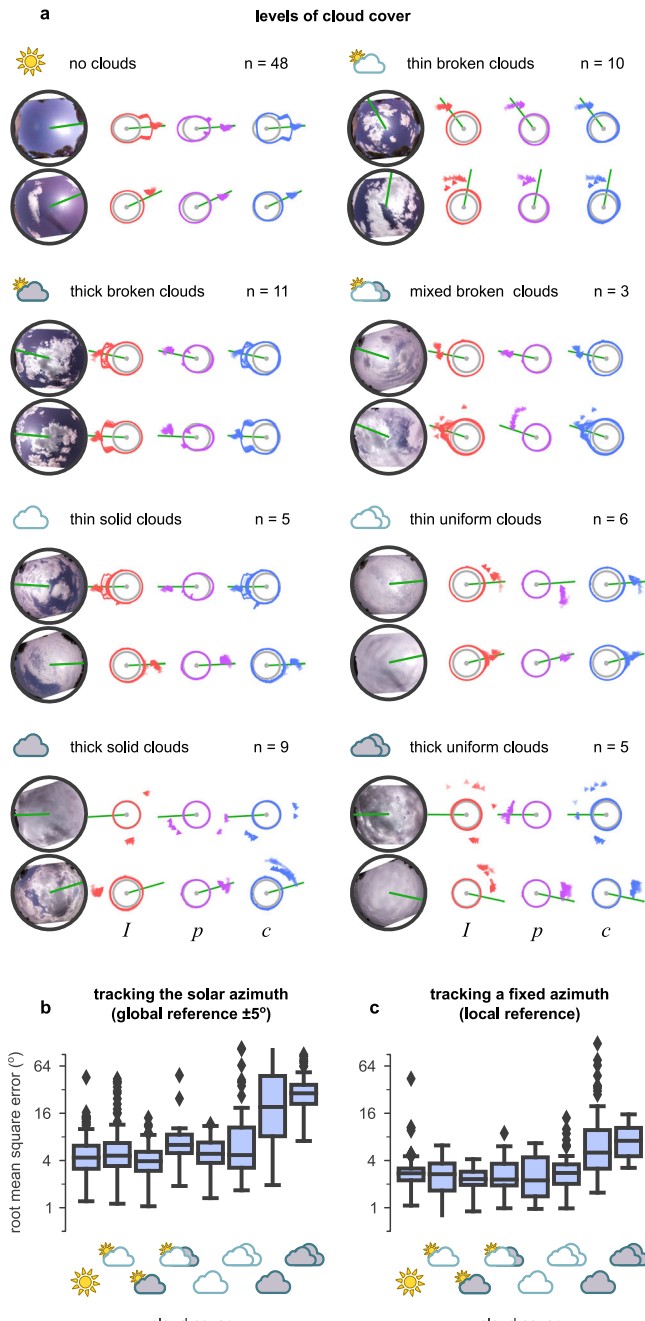

**Fig. 5 Sensor performance over cloudy skies. a** Examples of different levels of cloud cover in the dataset, in increasing difficulty: no clouds, thin broken clouds, thick broken clouds, mixed broken clouds, thin solid clouds, thin uniform clouds, thick solid clouds, and thick uniform clouds. Each block represents a different condition with two examples and reports the total number of available examples in the dataset (n). Left: fish-eye images of the sky for each example. Right: I, p, and c values for each example, the prediction of the solar azimuth using the different models (intensity, polarisation, and celestial compass). Each example shows the median (arrowhead) and quantiles (circle segment) of the 360 predictions produced (at homogeneously distributed orientations) during each of the twelve full rotations of the sensor. **b** Root mean squared error (RMSE) of estimating the solar azimuth (±5° error) using the celestial compass. **c** RMSE of tracking any fixed direction. Box-plot: centre line, median; box limits: upper and lower quartiles; whiskers: 1.5 × interquartile range; points: outliers. Vertical axes are in the log₂ scale. Ticks without labels denote (from bottom to top) 0°, 2°, 8°, 32°, and 90° RMSE.

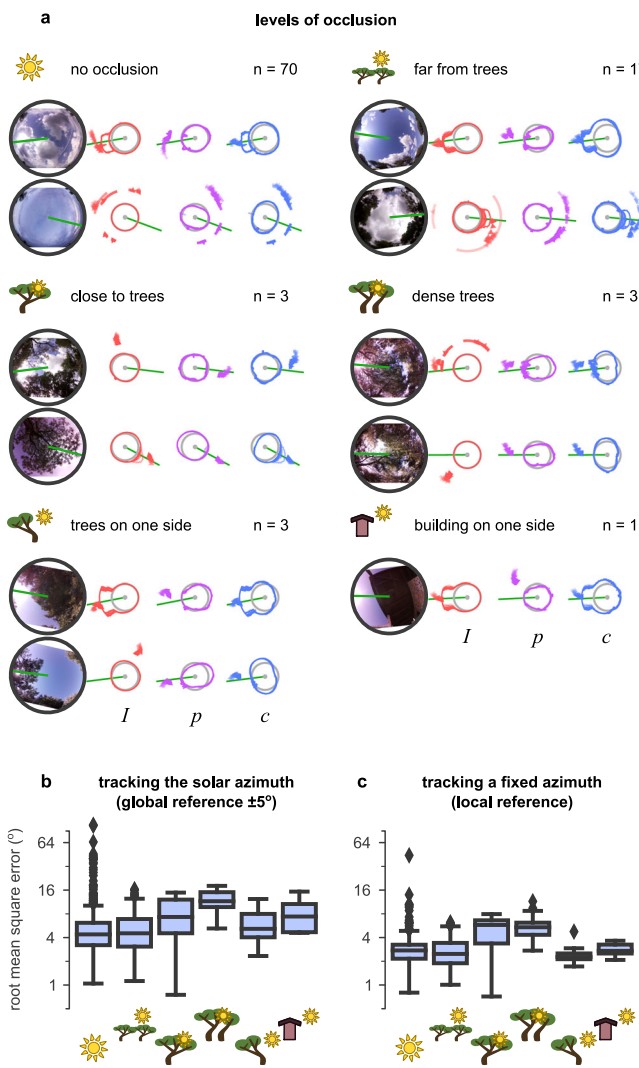

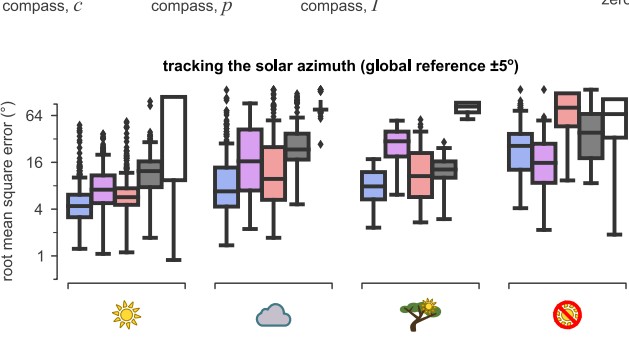

**Fig. 6 Sensor performance over occluded skies. a** Examples of different levels of occlusion in the dataset: no occlusion, occlusion based on sensor distance from trees (far or close), dense tree cover with openings, and trees or buildings on one side. Each block represents a different condition with up to two examples and reports the total number of available examples in the dataset (*n*). Left: fish-eye images of the sky for each example. Right: *I*, *p*, and *c* values in each example, the prediction of the solar azimuth using the different models (intensity, polarisation, and celestial compass). Each example shows the median (arrowhead) and quantiles (circle segment) of the 360 predictions produced (at homogeneously distributed orientations) during each of the twelve full rotations of the sensor. **b** Root mean squared error (RMSE) of estimating solar azimuth ( ± 5° error). **c** RMSE of tracking any fixed direction. Box-plot: centre line, median; box limits: upper and lower quartiles; whiskers: 1.5 × interquartile range; points: outliers. Vertical axes are in the log$_2$ scale.

**Alternative compass models.** We have noted several instances above in which the celestial compass ($c = I - p$) provides a better estimate than a compass using only intensity $I$ or polarisation $p$. Here we provide a more direct comparison between these three alternatives and two further models that use different methods to estimate the solar azimuth from polarisation measurements. Considering first the use of $c$ vs. $I$ or $p$ (Fig. 7), in nearly every condition, the celestial integration provided better estimates of the solar azimuth than the other two compasses. An exception was when the sun was completely hidden, in which case the

**Fig. 7 Performance of alternative models.** Boxes represent the distribution of root mean square error (RMSE) across the subset of the data where the sky was almost clear (sun), with thick clouds (cloud), with severe occlusions (tree), or where the sun was completely covered by clouds or canopies. Blue represents the celestial compass, purple represents the polarisation compass, red represents the intensity compass, dark grey represents the eigenvectors model, and white represents the four zeros model. We report the root mean square error (RMSE) of predicting the solar azimuth. The data shown are for solar elevations of at least 15°. Box-plot: centre line, median; box limits: upper and lower quartiles; whiskers: 1.5 × interquartile range; points: outliers.

polarisation compass was more accurate (Supplementary Fig. 2a, b and Supplementary Table 6). In the remaining cases, the intensity compass and celestial integration had a better performance than the polarisation compass alone. Increasing the number of PAAs (from eight to thirty-six) had almost no effect on the estimation of the intensity compass, but positively affected the estimations of the polarisation and celestial compasses in occluded skies. When tracking a fixed azimuth (local RMSE), the intensity and celestial compasses achieved the most accurate estimation of orientation (Supplementary Fig. 2c and Supplementary Table 6).

We also implemented two alternative models that can use the signals from our PAAs to extract the solar azimuth. In the first model, we calculated a unit vector for each PAA, which was directed perpendicularly to the AoP (as computed by the complete set of four photodiodes per PAA, at 0°, 45°, 90°, and 135°, and following the computations suggested by Zhao et al.[40]). However, the AoP ranges in [ − 90°, + 90°], which creates a 180° ambiguity of the direction. To resolve this, we assumed that all the unit vectors point towards the interior of the compass sensor (see Supplementary Fig. 3a). Then we followed the approach of Stürzl and Carey[15] and used the covariance of these vectors to calculate the eigenvector with the lowest eigenvalue, which should point towards the average direction of all these vectors, and we refer to this as the eigenvectors model. Numerous other celestial compasses used variations of this method[9–21]. Smith and Stewart proposed the second model[41,45,46] and suggested that, when rotating a tilted PAA, the polarisation responses ($p$) as calculated by equation (2) form a curve that takes both positive and negative values along the rotating axis and that this curve is zero at exactly four directions (see Supplementary Fig. 3b). The two directions that are the closest to each other should also be the closest to the solar azimuth. Thus the solar azimuth can be calculated as the mean direction of these two directions (described by the four zeros), and we refer to this as the four-zeros model. A detailed description of how we implemented these models can be found in Materials and Methods.

Figure 7 suggests that our proposed model outperformed the other two models described above in all the tested sky conditions. In the easiest scenario, where the compass was composed of 36 PAAs and used a local reference under almost clear sky conditions, our celestial compass model achieved (on average) 0.59° RMSE, while the eigenvector and four-zeros models achieved 2.70° and 70.38° respectively (Supplementary Fig. 2c and Supplementary Table 6). The four-zeros model was very fragile even for clear sky conditions, as any solar elevations higher than the PAA (>45°) produced only two zeros instead of four (Supplementary Fig. 3a—first example) and elevations below 15° produced evenly spaced zeros (Supplementary Fig. 3a,b—second example) resulting in a random direction choice. In most conditions, the eigenvector model performance was better than the four-zeros model but not as good as the celestial compass model. The performance of the eigenvector model was affected less by occlusions, but substantially affected by clouds, and more substantially when the sun was hidden completely. The main reason for this performance drop was the 180° ambiguity of the unit vectors (which were based on the AoP) and the assumption that they always face towards the interior of the compass. Impressive results reported previously for this method rely (instead of this assumption) on a calibration process to minimise the RMSE[11-21]. However, such a process does not seem biologically plausible and can also be expensive from a technical perspective.

## Discussion

Inspired by insect vision, we physically implemented the sensor design and computational model proposed by Gkanias et al.[29], verified that it can be used as a robust celestial compass sensor, and that even better estimations can be obtained by a simple integration of the skylight polarisation and intensity signals. The performance of the model was tested in various sky conditions, with different types of clouds and occlusions. Compared to the performance of two other compass models, it demonstrated superiority in its estimations of both the (global) solar azimuth and of a (local) point of reference.

Our computational model was also developed as an in-principle way to process the information from an array of PAAs. However, it is interesting to compare the algorithm to known processing pathways in the insect brain (illustrated in Fig. 8 for the *Drosophila melanogaster* fruit fly). The orthogonally placed polarisers of our PAAs imitate the microvilli of the photoreceptors in the dorsal rim ommatidia. Respectively, $I_{90}$ and $I_0$ values are analogous to the responses of R7 and R8 photoreceptor cells in the insect retina. Interestingly, R7 axons are inhibited by R8s, implementing the $I_{90} - I_0$ node in Fig. 2a[47,48]. Intensity information ($I$) is probably encoded by the Dm9 neurons, which collect R7 and R8 responses and feedback to the R7 axons. Polarisation information ($p$ values in our model) could then be encoded by the DmDRA1 interneurons and is projected to the anterior optic tubercle (AOTu) through a subset of medulla-tubercule (MeTu) neurons. This is because the R7-R8 difference encoded by the R7 axon is normalised by the Dm9 activity before connecting to the DmDRA1, which results in a response that is increased by R7 ($I_{90}$) and decreased by R8 activity ($I_0$)[47,49,50] or unpolarised light ($I$, feedback from Dm9)[48,51]. We predict that the final intensity information ($I$ value) is encoded by a different subset of MeTu neurons that receive direct input from Dm8 neurons (downstream of non-DRA R7 neurons that respond to unpolarised light)[48]. The responses of all these neurons are collected by the AOTu and communicated by the tubercular-bulb (TuBu) neurons to central brain regions. Note that the representation in the AOTu keeps the fan-like retinotopic structure of

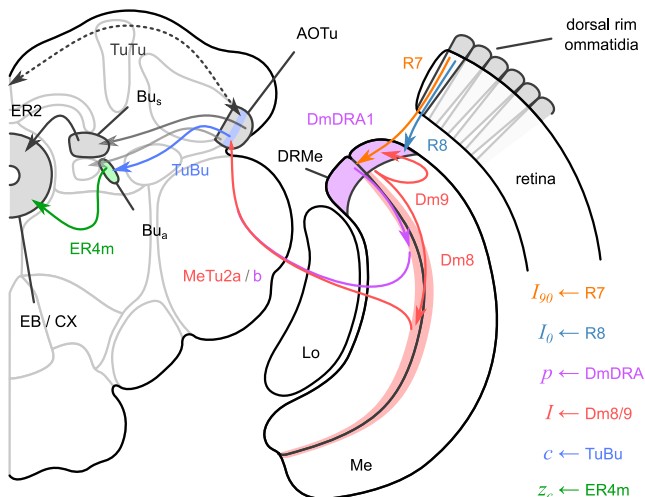

**Fig. 8 Pathway of the polarised light in the insect brain (here *Drosophila melanogaster*) and the suggested parallels to the processing in our model.** R7/8: retina neuron 7/8, Me: medulla, Lo: lobula, DmDRA1: distal medulla dorsal rim area neuron 1, Dm8/9: distal medulla neuron 8/9, MeTu: medulla-tubercule, AOTu: anterior optic tubercle, TuTu: tubercular-tubercle, TuBu: tubercular-bulb, Bu: bulb, Bu$_s$: superior bulb, Bu$_a$: anterior bulb, ER: ellipsoid-body from ring neurons, EB: ellipsoid body, CX: central complex. Adapted and modified from[51]; data from[48,51].

the DRA, but pooled and summarised in 10 columns per hemisphere[51]. Also, it establishes an inter-hemisphere communication[48,51-55], which we predict connects the two AOTus in a complete ring that represents the solar azimuth. Thus, our celestial integration values ($c$) should be analogous to the responses of the TuBu neurons and collectively represent a vector pointing towards the solar azimuth. However, these responses should also be affected by spectral and optic flow inputs[51], which are omitted from our model. The celestial compass of insects also corrects for the sun's movement during the day, which might happen through a circadian mechanism (modelled[29]) or synaptic plasticity. Therefore, downstream ring neurons roughly correspond to our $z_c$, but their responses might also reflect other inputs relevant to the sun's position, such as spectral cues and even circadian corrections to create a true compass.

Fruit flies (*D. melanogaster*) have been shown to integrate their absolute compass with self-motion in two different stages of processing: first in the AOTu (optic flow input), and later in the ellipsoid body (feedback from motor neurons and optical flow)[48,51]. In an interesting parallel, the use of Kalman filters[12,13,56] or recurrent neural networks[57,58] has been explored to improve the performance of some celestial compass sensors. This might therefore be a biologically plausible way to improve the performance of our compass. Insects also demonstrate colour opponency in the medulla, and this approach has been explored by Stürzl[16] showing an advantage in distinguishing sky from tree branches to improve celestial compass performance under canopies. Thus, adding photoreceptors that respond to different wavelengths could also improve the performance of our compass. Celestial compass sensors can also be used for navigation at night[44] when the moon replaces the sun and forms a similar pattern of AoP and DoP[5]. The photodiodes of our sensor responded in twilight (when the moon had a weak effect on the polarisation pattern of the sky; Supplementary Fig. 4) and revealed a potential for a nocturnal (as well as diurnal) function. This aligns with the abilities of insects[4,59] and, thus, a more

systematic test of the nocturnal abilities of our sensor in the future would be interesting.

Although our sensor might appear bulky (especially compared to camera approaches), we designed this prototype to be easily customised. In principle creating a miniaturised version seems straightforward, by integrating all the photodiodes onto one printed circuit board (PCB); and the first stages of optical processing to obtain $I$, $p$ and $c$ could happen onboard. An alternative could be a complementary metal-oxide-semiconductor (CMOS) with carefully tuned polarisers on the top to follow a fan-like arrangement (as in[60]) or the full dome as described in[29]. Nevertheless, to approach the size, speed, and efficiency of the insect brain, stretchable electronics[61,62], nanowire technology and photonic computations might be an interesting way forward.

## Materials and Methods

**Array of polarisation axis analysers.** Each PAA was made of 3D-printed housing, a PCB, 4 photodiodes that were sensitive to UV light (221–358 nm), and 4 linear polarisers that allowed UV light (280–450 nm) with AoP at (1) 45°, (2) 135°, (3) 0°, and (4) 90° with respect to the vertical axis of the PAA. The specific materials of the photodiodes, filters, and PCB parts are summarised in Supplementary Table 7.

Figure 9a shows the block diagram of the designed PCB for each PAA. It includes four photodiodes (SG01D-18, SGLUX) followed by their trans-impedance amplifiers (TIAs). A Schematic of the designed TIA is provided in Fig. 9b. Linear technology LTC6082 quad low offset (60 $\mu$V), bias current (1 pA), voltage noise (13 nv·Hz$^{-0.5}$) and current noise (0.5 fA·Hz$^{-0.5}$) CMOS operational amplifier (opamp) was used where the photodiode was operating in photoconductive mode. The output voltage, $V_{out}$ of the TIA was $I_{PD} \times R_F$, where $I_{PD}$ is the photodiode reverse

current which is proportional to the wavelength of the light and $R_F$ is the feedback resistor. A higher gain was achieved by using a 30 M$\Omega$ feedback resistor, which also helped to achieve a higher signal-to-noise ratio. A gain bandwidth product (GBWP) of 15 kHz was achieved by using a 6.8 pF feedback capacitor. This helped to improve the noise performance of the TIA and also provided a good transient response for the analogue-to-digital converter (ADC). The Texas Instruments ADS112C04 16-bit 4-channel precision delta-sigma ADC was selected. It was configured as differential mode ADC, where one channel was the input of the TIA output signal and another channel was fed with $V_{Bias}$ of 2.5 V. A multiplexer was used to select the differential input signals from a particular TIA and the bias voltage. A programmable gain amplifier was acting as a differential amplifier with a gain of 1. This configuration improved the dynamic range of the ADC, which was configured to run in normal mode, with a data rate of 45 Hz (samples per second). The ADC was set to continuous conversion mode and the input multiplexer was configured as necessary during operation. All other configurations used default options. Two ADS112C04 chips were used to accommodate 4 TIAs in the PCB, which were configured with different interintegrated circuit (I$^2$C) addresses. The power management unit was built with the linear technology ADP7118-2.5V and ADP7118-5V for the bias and power supply voltage respectively. A 6-pin JST-GH connection was used to connect the PCB of each PAA to the battery and processing unit. Figure 9c shows the result PCB.

As each PCB uses the same configuration, we included an I$^2$C multiplexer (TCA9548A, Adafruit) which allows sequential communication with each PCB. The I$^2$C multiplexer was mounted in a custom breakout board that used Grove Universal 4-pin connectors (Seeed Studio) to interface with connected I$^2$C PAAs, with a custom adaptor cable to link the 6-pin PCB

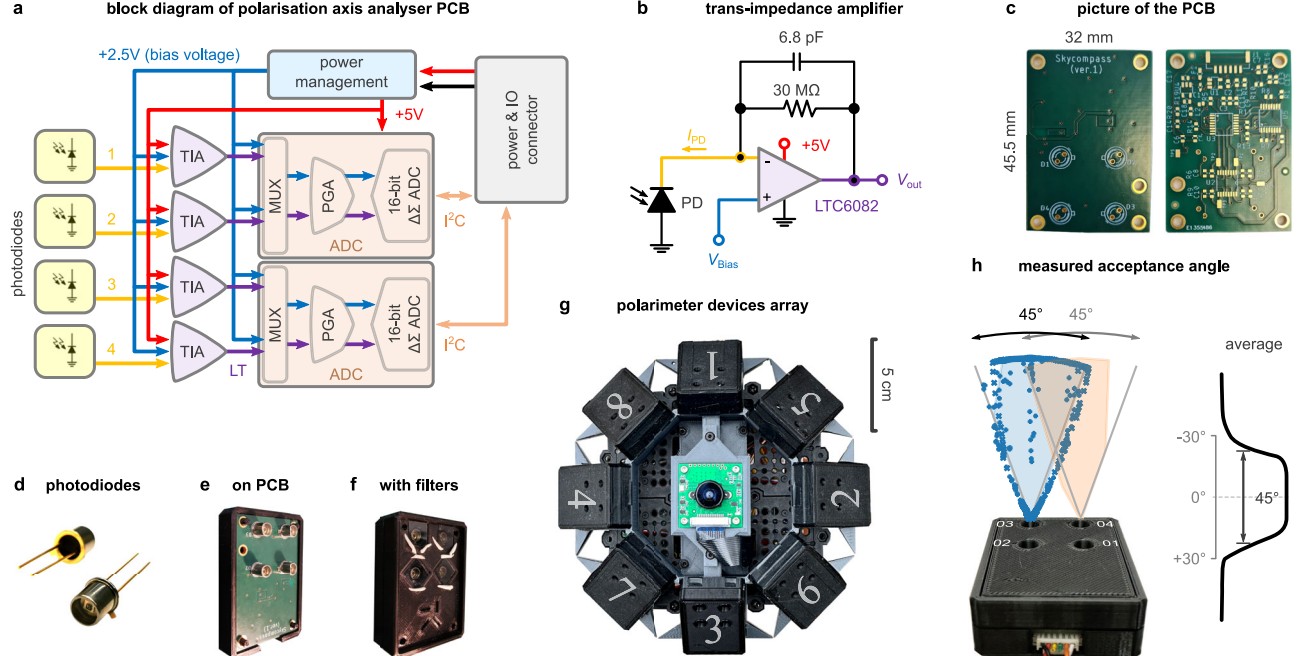

**Fig. 9 Assembly of the celestial compass sensor. a** Block diagram of the polarisation axis analyser (PAA) printed circuit board (PCB). Red lines represent positive voltage, blue lines are bias voltage, and the black line is ground; the rest are as indicated in the figure. TIA: trans-impedance amplifier, MUX: multiplexer, PGA: programmable gain amplifier, ADC: analogue-to-digital converter, I$^2$C: inter-integrated circuit. **b** Schematic of the TIA circuit. PD: photodiode. **c** Front (left) and back (right) images of the PCB. **d** Picture of the photodiodes. **e** Four photodiodes on the PCB, and (**f**) in the 3D printed casing with polarisation filters attached. **g** The array of PAAs and the camera mounted on the 3D-printed caddy. Numbers denote the identity of each PAA and the order in which their output was read. **h** The acceptance angle was measured using a torch; the average response distribution of the photodiodes was estimated by using data from photodiodes 3 and 4.

connector to the breakout board. To communicate with the multiplexer (and PCBs) we used the I$^2$C interface of Raspberry Pi and custom software using the Linux I$^2$C/system management bus subsystem (adapted from existing Arduino libraries).

Our sensor was 3D printed using polylactic acid (parts were printed on Ultimaker 2+ extended and Ultimaker S3 3D printers). Each PCB had its own housing, comprised of a base, mid, and top plate. The base and mid plates closed around the PCB. The mid plate has square insets over each photodiode, holding the polarisers (OUV2525, Knight Optical) and giving each photodiode its polarisation tuning (see Fig. 9d–f). A small amount of adhesive putty (e.g., Blu Tack, Bostik, or similar) was used to fix the polarisation filters in place (Fig. 9e). The PCB with filters and housing creates one PAA (Fig. 9f). Eight PAAs were constructed and arranged on a 3D-printed caddy which positioned the units with an elevation of 45° (Fig. 1h). The caddy was mounted to the robot using two different configurations (due to external constraints on the robot at the time). While in Sardinia, the caddy was mounted to the robot using a pillar at the rear of the robot base. While in South Africa, the sensor was mounted directly to the top of the robot base. We noted no difference in operation or performance using the different mounts—both sets of computer-aided design (CAD) files are available in Supplementary Data 1. The camera module was mounted in the centre of the caddy (Fig. 9g).

To measure the effective acceptance angle of each PAA, we placed it vertically on a protractor and flashed UV light from the side with a torch (Alonefire SV10, at 365 nm wavelength). We targeted a single photodiode at a time from different angles, and logged the apparent angle and recorded pulses from the photodiode—blue and orange shaded areas in Fig. 9h for photodiodes 3 and 4 respectively. We then recorded from the photodiode 3 while swinging the torch with roughly stable angular velocity and inferred the apparent angle of the torch using the carefully collected samples from before and linear interpolation (these are illustrated with blue crosses in Fig. 9h). There was a small deviation in the centre of the acceptance between photodiodes 3 and 4, probably because of a small deviation in their placement angle and was not intentional (Fig. 9h). The average acceptance angle was measured roughly to 45°; the photodiodes responded more weakly for up to 60°.

**The robot platform**. The robot was constructed using a Turtle-Bot3 Burger kit (ROBOTIS). The kit contained a Raspberry Pi 3B +, OpenCR1.0, two Dynamixel XL430-W250 actuators, and a light detection and ranging (LIDAR) subsystem which we did not use (full details are available in the Turtlebot e-manual provided by ROBOTIS). In addition, the kit contained several standardised structural plates and beams to construct the body of the robot as appropriate. The kit was augmented with a 3rd party Raspberry Pi Camera module (B0103, Arducam) which was mounted such that it pointed towards the zenith. Figure 1h illustrates the final appearance of the robot. The IMU is embedded in the OpenCR1.0 board included with the TurtleBot kit and interaction with it was facilitated by libraries provided by ROBOTIS for the OpenCR1.0 platform.

The Turtlebot was controlled via the robot operating system (ROS). A ROS master node ran on a host laptop (ROS Noetic, Ubuntu 20.04) which generated a Wi-Fi hotspot. The Turtlebot (ROS Kinetic, Raspbian 9 stretch) was connected to this Wi-Fi hotspot as soon as it was available and was configured to view the host laptop IP as the ROS master. Thus, ROS nodes running on the Turtlebot and host laptop could communicate. For all data presented, the laptop ran only the recording routine (see below). On board, we ran nodes to read from the sensor and camera (as

well as all those concerned with the basic operation of the Turtlebot). On the host laptop, we ran the `roscore` and recording routine. All data were recorded using the `rosbag` C++ application programming interface (API). All interaction with the IMU was performed via ROS.

**Data collection**. Data was collected in three locations: Sardinia (Italy; 39.258648N, 8.440184E) in May 2022, Vryburg (South Africa; -26.398643N, 24.327144E) from 10 to 16 November 2022, and Bela Bela (South Africa; -24.714872N, 27.918972E) from 16 to 29 November 2022. During each recording session, the robot was placed on a smooth level surface facing approximately north. The robot would then record statically for five seconds, rotate slowly through 360°, and then record statically for five seconds. On subsequent rotations, the robot would correct using the IMU for any over-rotation before starting to record. The IMU was periodically re-initialised to prevent noticeable drifts over the course of a recording session. At the beginning of each recording session, a time-stamped sky photo was captured using the onboard camera. Recording sessions lasted around 12-15 minutes on average and consisted of 12 rotations under the condition of interest (different solar elevations, cloud cover, or occlusions).

**Compass models**. The output of the photodiodes (voltage pulses scaled down by a factor of $11 \cdot 10^3$ and clipped to [0, 1]) was used from our model to estimate the solar azimuth, which represented the compass direction. Using this prediction, an ephemeris function (describing the course of the sun during the day) and the coordinated universal time (UTC) we can accurately estimate the true north as well as an approximation to the coordinates of the sensor on Earth. Equation (1), (2), (3), (4), and (5) describe how our model transforms the photodiode pulses into a compass direction. For comparison, we implemented two additional compass models: the eigenvectors[16,40] and the four-zeros[41]. All the models were implemented in Python 3.

For the eigenvector model, we first needed to calculate the AoP ($a$) and DoP ($d$) from the photodiode responses. We calculated these following Zhao et al.[40], whose PAAs were similar to ours,

$$a_n = \frac{1}{2} \arctan \frac{r_{2,n} - r_{1,n} + r_{1,n} r_{2,n} - 1}{r_{1,n} - r_{2,n} + r_{1,n} r_{2,n} - 1} + \phi_n, \qquad (6)$$

$$d_n = \frac{r_{1,n} - 1}{(r_{1,n} + 1) \cos(2a_n)}, \qquad (7)$$

$$r_{1,n} = \frac{I_{0,n}}{I_{90,n}}, \qquad r_{2,n} = \frac{I_{45,n}}{I_{135,n}}, \qquad (8)$$

where $\phi_n = 2\pi(n-1)/N$ is the azimuth angle of the respective PAA, and $n$ is its identity as illustrated in Fig. 9g. The respected elevation of each PAA is always 45° ($\pi/4$). The polarisation vector associated with each PAA is perpendicular to the respective AoP and points towards the inside of our sensor,

$$\mathbf{p}_n = [\sin(a_n + \pi) \qquad \cos(a_n + \pi)]. \qquad (9)$$

Weighting each of these vectors with the respective DoP ($d_n$) did not affect any of our results. The covariance matrix of these vectors was calculated as

$$\mathbf{C} = \mathbf{P} \cdot \mathbf{P}^T, \qquad (10)$$

and its eigenvectors represent the principal facing axes. The eigenvector with the highest eigenvalue ($\hat{\mathbf{e}}$) points towards the solar azimuth[16]. We calculated the eigenvectors and eigenvalues of the covariance using the NumPy package[63], and transformed this eigenvector with the highest eigenvalue into a complex

number for consistency with the other two models,

$$z_{ev} = \hat{e}_2 + i\,\hat{e}_1, \tag{11}$$

while the estimated solar azimuth ($\alpha_{ev}$) can be calculated using equation (5). Three examples of the AoP, **p** vectors, and solar azimuth estimations of this model are illustrated in Supplementary Fig. 3a. For the four-zeros model, we used as input the polarisation responses ($p_n$) of $N$ PAAs, which were computed using equation (2). For this model, we needed to fit a curve on the responses, which could be described by the first three Fourier coefficients of the DFT. However, we were interested in centring the curve to zero, so the zero-coefficient can be omitted,

$$z_1 = \frac{2}{N}\sum_{n=1}^{N} p_n\,e^{i2\pi(n-1)/N}, \tag{12}$$

$$z_2 = \frac{2}{N}\sum_{n=1}^{N} p_n\,e^{i4\pi(n-1)/N}. \tag{13}$$

The magnitude of these coefficients is $\rho_k = ||z_k||$ ($k \in \{1, 2\}$), and their angle ($\alpha_k$) is given by equation (5). The function describing the curve of the responses is then

$$f(\theta) = \rho_1\,\cos(\alpha_1 - \theta) + \rho_2\,\cos(\alpha_2 - 2\,\theta), \tag{14}$$

which is plotted in Supplementary Fig. 3b in black, and its derivative is

$$\frac{df}{d\theta} = \rho_1\,\sin(\alpha_1 - \theta) + 2\,\rho_2\,\sin(\alpha_2 - 2\,\theta). \tag{15}$$

We used these equations as input to the Newton-Raphson optimiser (from the SciPy package[64]) to estimate the four solutions of equation (14). We ran the optimation four times with different initialisations that were homogeneously distributed around the angle of the second coefficient (i.e., $\theta_{init} = \alpha_2/2 \pm \pi/2 \pm \pi/4$). This ensured that the optimisation starts roughly at the correct position and falls in the correct local solution. Examples of these solutions are plotted in Supplementary Fig. 3b (red dots), demonstrating the correctness of the method so far. Next, the angles of the four zeros were normalised in $[0, 2\pi)$ and sorted. The absolute difference between the consecutive solutions was calculated, and the pair of solutions with the lowest difference was identified,

$$\delta_m = ||(\theta_{(m+1\,\mathrm{mod}\,4)} - \theta_m + \pi)\,\mathrm{mod}\,2\pi - \pi||, \tag{16}$$

$$\hat{m}_1 = \arg\min_{m=1}^{4} \delta_m, \tag{17}$$

$$\hat{m}_2 = \hat{m}_1 + 1\,\mathrm{mod}\,4 \tag{18}$$

where mod denotes and modulo operation, and arg min finds the identity that represents the minimum value of $\delta_m$. The estimated solar azimuth is then calculated,

$$\alpha_{fz} = \theta_{\hat{m}_1} + \frac{\delta_{\hat{m}_2}}{2}. \tag{19}$$

**Performance evaluation**. We evaluated the performance of the models by using the RMSE as a global and local error measurement across different orientations of the sensor in the same sky condition and rotation. The RMSE for a specific model, sky condition, and rotation was calculated as

$$\mathrm{RMSE}_{model} = \sqrt{\frac{1}{360}\sum_{s=1}^{360} \epsilon^2_{model,s}}, \tag{20}$$

where $\epsilon_{model,s}$ represents the (global or local) error of the model, and $s \in \{1, \ldots, 360\}$ are the homogeneously distributed

orientations of the sensor during a rotation with respect to the starting orientation.

The global error represents the overall deviation of the estimated solar azimuth from the real solar azimuth during a rotation ($\pm 5°$, due to the approximate initialisation towards the north),

$$\epsilon^G_{model,s} = (\alpha_{model,s} - \hat{\alpha}_s + \pi)\,\mathrm{mod}\,(2\pi) - \pi, \tag{21}$$

where $\alpha_{model,s}$ is the prediction of the model for the solar azimuth (with respect to the front of the sensor) when the sensor was in orientation $s$, and $\hat{\alpha}_s$ is the respective true solar azimuth ($\pm 5°$).

The local RMSE is the overall deviation of the estimated solar azimuth from the average estimated solar azimuth, and it represents the consistency of the predictions of the sensor,

$$\epsilon^L_{model,s} = (\alpha_{model,s} - \bar{\alpha}_{model} + \pi)\,\mathrm{mod}\,(2\pi) - \pi, \tag{22}$$

where $\bar{\alpha}_{model}$ is the average prediction of the model for the solar azimuth across the different tested orientations ($s$).

### Data availability
The data regarding the sensor design (CAD files) are provided in Supplementary Data 1. The raw and summarised data that were collected using the robot are publicly available in DataShare with identifier https://doi.org/10.7488/ds/6106.

### Code availability
The code used for the robot, data collection, analysis and plots is available on GitHub. Code for robot and data collection is available by the authors upon request. Code for the data analysis and generation of plots is publically available in zenodo with identifier https://doi.org/10.5281/zenodo.8393056.

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

## Acknowledgements
This work was financially supported in part by the European Union, Horizon Europe, (Project 101046790, InsectNeuroNano), the European Research Council (817535, Ultimate-COMPASS), and the UKRI Engineering and Physical Sciences Research Council (Grant No. EP/R513209/1, EP/M008479/1, and Impact Acceleration Account PIV068). We thank Marie Dacke from Lund University for facilitating field testing by providing travel and access to the field sites used for data collection. Thanks to the members of the Insect Robotics Group from the University of Edinburgh for commenting on earlier versions of the manuscript. Also, thanks to the InsectNeuroNano team for comments on the results of this work.

## Author contributions
E.G.: conceptualisation, formal analysis, investigation, methodology, project administration, software, validation, visualisation, writing—original draft, writing—review & editing. R.M.: conceptualisation, data collection, investigation, methodology, software, hardware, visualisation, writing—original draft, writing—review & editing. J.S.: conceptualisation, funding acquisition, investigation, hardware. Sadeque Reza Khan: conceptualisation, hardware, visualisation, writing—original draft, writing—review &

editing. S.M.: conceptualisation, resources, supervision, writing—review & editing. Barbara Webb: conceptualisation, funding acquisition, investigation, methodology, project administration, resources, supervision, validation, writing—review & editing.

## Competing interests

The authors declare no competing interests.
