## [Peer Review File · Communications Engineering]

Reviewers' comments:

Reviewer #1 (Remarks to the Author):

This manuscript presents a sensor that mimics critical characteristics of insect eyes, especially the fan-like setup of polarised light receptors in their upper rim zone. The sensor is composed of a circular array of merely eight polarimeter units to evaluate the skylight arriving from multiple directions. Given that the sensor's configuration corresponds to the pattern of polarised light in the sky, a straightforward computation that amalgamates data from across the devices spatially allows precise determination of the sun's location. During comprehensive validation involving multiple obstructions from trees and buildings, along with a variety of atmospheric and weather conditions, they demonstrated exceptional performance in comparison to other, computationally more intricate methods. This manuscript is well-written and almost ready to publish. I have a few comments for improving the quality of this manuscript.

- The authors used eight individual devices for mimicking dorsal rim ommatidia. Even the robot should have fish-eye cameras, which is pretty much bulkier than the natural compound eyes. The authors should mention the scaling issue of such compound eye vision system. Several researchers are focusing on flexible/stretchable electronics for reducing the modules. Recent approach even enables a panoramic visual field.

- In Figure 3a, the root mean square error has broader ranges as the number of devices increase, which block the accurate tracking the sun location. The authors could comment on that.

- The reviewer is wondering if this research can be further extended to detect circular polarization from the object.

Reviewer #2 (Remarks to the Author):

This is an interesting work proposed by Gkaniyas et al. about the implementation of a celestial compass composed of only 8 polarimeter devices. Each polarimeter integrates 4 photodiodes sensitive to UV and 4 polarized filters oriented to 4 different directions. But only two of the four directions are used for the study. The study includes simulated and real data obtained in real condition outdoor under various cloud covers and even at dawn and dusk. The hardware implementation of the compass is clearly described and the processing of the visual informations based on the use of circular mean is also well described and interesting. However, the main critical point that can be pointed is the accuracy of the compass, which is

about 4° even in a clear sky condition (i.e., the most favorable condition).

Authors argue in the abstract that their device allows an accurate extraction of the sun's position : 4 degrees is not accurate and not very acceptable in terms of navigation, especially compared to the state of the art with similar non-imaging sensors. By the way the following papers should have been cited and their performances fairly compared with those of the study:

- Accuracy of 0.2° : Wang et al. (2018) Orthogonal vector algorithm to obtain the solar vector using the single-scattering Rayleigh model
- Accuracy of 0.2° (sun tracking) : Zhang (2021) : Bionic integrated positioning mechanism based on bioinspired polarization compass and inertial navigation system
- Accuracy of 0.2° : Wang et al. (2014) Empirical corroboration of an earlier theoretical resolution to the UV paradox of insect polarized skylight orientation

I think that the following paper measurements under the canopy should be cited: Dupeyroux et al. (2019) An ant-inspired celestial compass applied to autonomous outdoor robot navigation

In addition, I strongly suggest authors to calibrate the response of each photodiode with their polarized filter. The Malus law drawn in figure 1f must be calibrated carefully because the gain of the photodiodes is not the same for each photodiode. I recommend to look at the calibration method proposed by Du al. (2019) Multiple disturbance analysis and calibration of an inspired polarization sensor. I think that the relatively bad accuracy of the compass is due to a lack of calibration.

Authors claim that the processing method they used is bio-inspired but the use of circular mean by animals remains to be shown.

In figure 3a, I do not understand why the higher the number of devices, the larger the disparity? In figure 4b, I do not understand why the accuracy is so bad at very low elevation of the sun where the intensity of the polarized light is maximum at the zenith.

Authors compared the performances of their compass to other methods proposed by Smith et al., which are not really the state of the art and which seem to feature a very bad accuracy. I recommend authors to compare their method with other standard method based on the Stokes formalism.

Page 7 line 393 : authors compared errors to a compass with 36 polarimeters which has only been simulated and not tested experimentally. This kind of comparison is not very fair and participates to blur the real performances of the device.

For all these reasons, I do not recommend this paper for a publication in Communications Engineering journal. The method is interesting but I do not understand why authors finally did not use the four directions available for each polarimetric device and thus apply the classical Stokes formalism that could certainly lead to better performances.

Reviewer #3 (Remarks to the Author):

The paper by Gkaniyas et al. describes the design and testing of a robot that uses sensors inspired by insect photoreceptors to detect the polarization pattern and intensity gradients in the sky for spatial orientation. The robot shows superb sun compass performance even under unfavorable conditions such as different clouding or under canopies and thus impressively illustrates the power of insect brain mechanisms in sky compass navigation. The design has several advantages over previous technical solutions and is therefore an important contribution to the field of autonomously navigating vehicles. I have several comments that might help to further improve the manuscript.

1. You use the term “polarimeter” for your polarization axis analyzers. According to Wikipedia, a polarimeter is a device to measure the angle of rotation caused by passing linearly polarized light through an optically active substance, which is clearly not how the term is used here. You might either introduce a new definition for the term or use an alternative term such as “polarization axis analyzer” or “polarization angle analyzer”.

2. Throughout the manuscript it seems to me that you use the terms “solar azimuth”, “solar position” and “solar meridian” somewhat interchangeably, but each of these terms describes different things. The solar position is defined by a horizontal component, termed solar azimuth and a vertical component, the solar elevation. The solar meridian, on the other hand, is the great circle in the sky passing through the sun and the zenith. It might be helpful to add a figure describing these terms. What your robot is apparently capable of doing is to navigate according to solar azimuth, but in the summary you say that it “allows accurate extraction of the sun’s position” (which would include its elevation). Is that so? If so, how precise is it in this respect?

3. Line 22: “anti-solar meridian”. This is not exactly correct. The degree of polarization is strongest along a circle in the sky that is 90° away from the sun.

4. Lines 42-44: These are very general statements that may be true for many insects but clearly not for all. Extreme cases are e.g. cave insects which are blind. Likewise, dorsal rim areas have not been reported in all insect orders (see Labhart and Meyer 1999; *Microscopy Research and Technique* 47:368). So perhaps modify the statements by saying “...many insects..”

5. Fig. 1 legend: Many images here seem to be taken from other sources, such as Fig. 1b,c,f and g. Please acknowledge these sources in the figure legend.

6. Line 139: ...that points towards the solar meridian...” Is this true? The solar meridian is a great circle passing through the sun and the zenith. Do you rather mean “position of the sun” or “solar azimuth”?

7. Figure 8: there is no reference in the figure legend acknowledging the source of this image or the origin of data in this image.

8. Line 422: why not use the simpler term “medulla-tubercle (MeTu) neurons.”

9. Line 433: likewise, “tubercle-bulb (TuBu) neurons” would be a bit simpler.

10. Line 452 following: This paragraph is based on evidence from the fly *Drosophila*. Therefore, do not use the general statement on line 451 "Insects have been shown..." Specifically refer to *Drosophila*.

11. Line 725, 727: The name of the author is "Lord Rayleigh" F.R.S. stands for "Fellow of the Royal Society" which is not part of the author's name.

12. Line 731: The initials of this author are J.W. (not H.J. as indicated here).

Response to comments on manuscript COMMS-23-0250-T

Reviewer #1:

This manuscript presents a sensor that mimics critical characteristics of insect eyes, especially the fan-like setup of polarised light receptors in their upper rim zone. The sensor is composed of a circular array of merely eight polarimeter units to evaluate the skylight arriving from multiple directions. Given that the sensor's configuration corresponds to the pattern of polarised light in the sky, a straightforward computation that amalgamates data from across the devices spatially allows precise determination of the sun's location. During comprehensive validation involving multiple obstructions from trees and buildings, along with a variety of atmospheric and weather conditions, they demonstrated exceptional performance in comparison to other, computationally more intricate methods. This manuscript is well-written and almost ready to publish. I have a few comments for improving the quality of this manuscript.

1. The authors used eight individual devices for mimicking dorsal rim ommatidia. Even the robot should have fish-eye cameras, which is pretty much bulkier than the natural compound eyes. The authors should mention the scaling issue of such compound eye vision system. Several researchers are focusing on flexible/stretchable electronics for reducing the modules. Recent approach even enables a panoramic visual field.

Ans: We thank the reviewer for the excellent suggestion. We have now modified the last paragraph of our discussion to address this comment:

Lines 455-468: *“Although our sensor might appear bulky (especially compared to camera approaches), we designed this prototype to be easily customised. In principle creating a miniaturised version seems straightforward, by integrating all the photodiodes onto one PCB; and the first stages of optical processing to obtain l , p and c could happen onboard. An alternative could be a CMOS with carefully tuned polarisers on the top to follow a fan-like arrangement (as in [58]) or the full dome as described in [27]. Nevertheless, to approach the size, speed, and efficiency of the insect brain, stretchable electronics [59, 60], nanowire technology and photonic computations might be an interesting way forward.”*

2. In Figure 3a, the root mean square error has broader ranges as the number of devices increase, which block the accurate tracking the sun location. The authors could comment on that.

Ans: The spread in Fig. 3a actually varies very little (STD = 4.5° - 5°) except of when we use 3 PAAs, where STD = 21.7°. Both axes on this plot are log-scale (horizontal is log₆ and vertical is log₂), which makes the spread of the data-points appear wider for smaller RMSEs than

for larger ones. As the RMSE in Fig. 3a drops, the spread naturally appears larger, but this is just an illusion caused by the visualisation (see illustrations above for double log-scale, linear horizontal axis, and double linear axes; from left to right). We've added the below lines at the end of Fig. 3 caption to clarify this.

*“Although the spread appears to be increasing with higher spatial resolution, this is an illusion of the logarithmic scale of the vertical axis. Standard deviation (SD) in **a** is 21.77° for 3 analysers and it falls in the range 4.41° to 5.15° from 4 to 60 analysers (randomly distributed). The SD in **b** is 21.92° for 3 analysers and it falls in the range from 2.82° (4 analysers) to 0.48° (60 analysers; decreasing for higher spatial resolutions) for the rest of the cases.”*

3. The reviewer is wondering if this research can be further extended to detect circular polarization from the object.

Ans: For the analysis of circular polarised light, we need to calculate the fourth Stokes parameter, which needs polarisers placed in series. Our sensor could be extended with another 2 photodiodes, placed behind a linear polariser at 0° followed by another two polarisers at 45° and 135° respectively. This would allow only clockwise or anti-clockwise circular polarised light to pass respectively, and the normalised difference of the photodiode pulses could give us the fourth Stokes parameter. However, the PAAs of our sensor point in different directions that merely overlap, and a different design would be needed to focus the PAAs on a single object. The ommatidia in the DRA of insects analyse the light in two orthogonal polarisation axes, so it seems unlikely that they can detect circular polarisation. However, some invertebrates have developed ommatidia (but not in the DRA) that can detect circular polarisation; for example, the stomatopods (shrimp mantis) and jewel scarab beetles (*Chrysina gloriosa*). To our knowledge, these animals do not use circular polarisation to estimate the solar azimuth.

Reviewer #2:

This is an interesting work proposed by Gkaniyas et al. about the implementation of a celestial compass composed of only 8 polarimeter devices. Each polarimeter integrates 4 photodiodes sensitive to UV and 4 polarized filters oriented to 4 different directions. But only two of the four directions are used for the study. The study includes simulated and real data obtained in real condition outdoor under various cloud covers and even at dawn and dusk. The hardware implementation of the compass is clearly described and the processing of the visual information based on the use of circular mean is also well described and interesting. However, the main critical point that can be pointed is the accuracy of the compass, which is about 4° even in a clear sky condition (i.e., the most favourable condition).

1. Authors argue in the abstract that their device allows an accurate extraction of the sun's position : 4 degrees is not accurate and not very acceptable in terms of navigation, especially compared to the state of the art with similar non-imaging sensors.

Ans: We thank the reviewer for valuable comments. We assume the reviewer is referring to the global error, as shown in Fig. 3a (i.e., estimate of sun position from one reading from the sensor) when using 8 devices (name changed to polarisation axis analysers, PAAs, after the revision). As we hoped to make clear in the paper the global error is subject to an additional initialisation error (+/-5°) due to constraints in our data collection procedure, setting a lower bound on the accuracy. A more fair comparison to the reported accuracy in state of the art would be the result with 60 PAAs and local error (Fig. 3b), as this is what is usually reported by others. In this case our median RMSE is 0.43°, which should be acceptable for navigation. However, we appreciate the reviewer's point and have modulated our respective claims in the abstract by changing:

"... allows an accurate extraction of the sun's position." to *"... can estimate the solar azimuth."*, and

"... our sensor exhibited superior performance compared to alternative (and computationally more complex) methods." to *"... our method exhibits superior performance compared to alternative (and computationally more complex) algorithms when using the same sensor design"*.

See also the further discussion of the choice of comparison algorithms below

2. By the way the following papers should have been cited and their performances fairly compared with those of the study:
 - a. Accuracy of 0.2° : Wang et al. (2018) Orthogonal vector algorithm to obtain the solar vector using the single-scattering Rayleigh model

Ans: This is an excellent suggestion. We cited an older version of this work (Wang et al (2015) DOI: 10.1038/srep09725), but in this updated version they have 5 PAA-like devices instead of 2, pointing towards 4 different directions at 45° elevation and also at the zenith. Each device had 6 photodiodes (30 in total), and they compare

different algorithms for extracting the solar azimuth. This work is now cited in our manuscript. However, the performance is not clearly reported and thus hard to compare to our performance. Specifically, this work (as well as others; see for example Lambrinos et al., *Adapt. Behav.*, 1997; *CVMR*, 1998; *RAS*, 2000; Carey and Sturzl, *ICCVW*, 2011; Han et al., *Sensors*, 2017; Wang et al., *IEEE Sensors*, 2017) reports the mean error (ME), as opposed to the mean absolute error (MAE) or root mean square error (RMSE) that would be comparable to our results. ME provides a more optimistic evaluation of the sensor, as positive and negative errors compensate for each other and it does not reflect the expected error for an individual measurement, which we consider to be the accuracy measure of interest.

- b. Accuracy of 0.2° (sun tracking) : Zhang (2021) : Bionic integrated positioning mechanism based on bioinspired polarization compass and inertial navigation system

Ans: The sensor presented in this paper is similar to Du et al. (2020, *IEEE Sensors*) and Liu et al. (2021, *Bioinspiration & Biomimetics*), both of which were cited. All these works report the performance of the fusion of a polarisation sensor with an inertial navigation system (INS) using different Kalman filters techniques. The reported accuracy on the paper suggested by the reviewer relates to the estimation of the solar elevation (not the azimuth), as they try to use it along with an ephemeris function to estimate the solar drift during the course of the day. Thus, we believe it is less relevant to cite this paper than the ones we already do.

- c. Accuracy of 0.2° : Wang et al. (2014) Empirical corroboration of an earlier theoretical resolution to the UV paradox of insect polarized skylight orientation

Ans: Another excellent suggestion. This work is very similar to the Sahabot work from Lambrinos et al., but they test the sensor in different wavelengths and in cloudy skies. This work is cited now in our manuscript. This sensor rotates the photodiodes to scan the environment (similarly to our interpolation method for up to 60 PAAs = 60 samples) before estimating the overall AoP at the zenith. They then use the knowledge that the AoP at the zenith is perpendicular to the solar azimuth, which gives two solutions and introduces ambiguity (similarly to Lambrinos et al., Dupeyroux et al, and others). Our paper already (in the introduction, provides a comparison to this general approach, highlighting the novelty of our sensor is that it does not use an intermediate step of computing the AoP, but it uses the sensor's geometry as a 'matched filter' to calculate the deviation of the polarisation pattern in the sky from its dorsal-most point, and hence does not need an additional measure to resolve ambiguity.

- d. I think that the following paper measurements under the canopy should be cited: Dupeyroux et al. (2019) An ant-inspired celestial compass applied to autonomous outdoor robot navigation

Ans: We thank the reviewer for also pointing to this work. This work tests the performance of estimating the AoP at the zenith in clear sky, under canopies, and even under water, which is impressive. We now replaced the citation of the same

sensor (Dupeyroux et al., 2019, Science Robotics) to this one (Dupeyroux et al., 2019, RAS) as this also tests the sensor under canopies and it is more relevant to our work.

3. In addition, I strongly suggest authors to calibrate the response of each photodiode with their polarized filter. The Malus law drawn in figure 1f must be calibrated carefully because the gain of the photodiodes is not the same for each photodiode. I recommend to look at the calibration method proposed by Du al. (2019) Multiple disturbance analysis and calibration of an inspired polarization sensor. I think that the relatively bad accuracy of the compass is due to a lack of calibration.

Ans: We thank the reviewer for the kind suggestion. While we agree it would be interesting to see how calibration of the sensor could improve performance, we do not have easy access to the required facilities, such as those used in Du et al. (2019) to undertake such a calibration as part of this paper revision. We also note that obtaining consistent performance despite the lack of calibration could be considered an advantage of our approach, and enhances the biological plausibility as is difficult to imagine how an insect could calibrate for variable sensor characteristics in its eye.

4. Authors claim that the processing method they used is bio-inspired but the use of circular mean by animals remains to be shown.

Ans: We respectfully disagree with the reviewer in this matter. It is now quite established that insects can represent 2D vectors and make calculations with them using sinusoidal patterns of activity or sinusoidal connectivity patterns in a population level (Stone et al., 2017; Green et al., 2017; Lyu et al., 2021). This evidence exists for circuits related to their central complex, which is exactly where we suggest that our calculations take place. In Gkaniyas et al. (2019), we suggested a way that the complex number calculations in equation (4) can be implemented using sinusoidal patterns of neural responses and synaptic weights.

5. In figure 3a, I do not understand why the higher the number of devices, the larger the disparity?

Ans: The spread in Fig. 3a actually varies very little ($STD = 4.5^\circ - 5^\circ$) except of when we use 3 PAAs, where $STD = 21.7^\circ$. Both axes on this plot are log-scale (horizontal is \log_6 and vertical is \log_2), which makes the spread of the data-points appear wider for smaller RMSEs than

for larger ones. As the RMSE in Fig. 3a drops, the spread naturally appears larger, but this is just an illusion caused by the visualisation (see illustrations above for double log-scale, linear horizontal axis, and double linear axes; from left to right). We've added the below lines at the end of Fig. 3 caption to clarify this.

*“Although the spread appears to be increasing with higher spatial resolution, this is an illusion of the logarithmic scale of the vertical axis. Standard deviation (SD) in **a** is 21.77° for 3 analysers and it falls in the range 4.41° to 5.15° from 4 to 60 analysers (randomly distributed). The SD in **b** is 21.92° for 3 analysers and it falls in the range from 2.82° (4 analysers) to 0.48° (60 analysers; decreasing for higher spatial resolutions) for the rest of the cases.”*

6. In figure 4b, I do not understand why the accuracy is so bad at very low elevation of the sun where the intensity of the polarized light is maximum at the zenith.

Ans: Unlike previous sensor designs, our sensor does not compute orientation by analysing the AoP at the zenith. Rather, it (effectively) determines the azimuth where the degree of polarisation in the sky is highest, which should be opposite to the sun’s azimuth. When, at low sun elevations, the intensity of the polarised light is strongest near the zenith, the azimuth of this point (of highest polarisation) has maximum ambiguity. To some extent this can be compensated by use of the intensity pattern, which should, correspondingly, be least ambiguous when the sun is near the horizon. However, we found that intensity patterns (and polarisation patterns) in this situation were sometimes affected by the moon, leading to poor accuracy overall.

Lines 443-447: “The photodiodes of our sensor responded in twilight (when the moon had a weak effect on the polarisation pattern of the sky; Supplementary Fig. S4) and revealed a potential for a nocturnal (as well as diurnal) function.”

7. Authors compared the performances of their compass to other methods proposed by Smith et al., which are not really the state of the art and which seem to feature a very bad accuracy. I recommend authors to compare their method with other standard method based on the Stokes formalism.

Ans: We chose the method from Smith because it is the only one based on a similar understanding of the insect eye: that polarisation sensors arranged (or rotated) in a circle at a consistent angle away from the zenith could be used to detect the sun direction without ambiguity. We agree that this method is not state of the art (and clearly performs poorly) but we did also include comparison to the eigenvectors method, which has been reported quite widely in the literature as a successful approach, e.g. in a variant by Wang et al. (2018, a suggested reference from the reviewer above). These methods rely on the accurate calibration of the sensor’s parameters to avoid the more simplistic assumption that we used regarding the p-vector direction. The AoP and DoP per PAAs seems correct as calculated by this method (see Supplementary Fig. S3a); however, the performance of this method using the same number of samples as our method (for both 8 and 36 PAAs) seems slightly worse (see Supplementary Fig. S2 and Supplementary Table S6). We note that a number of other state of the art methods cannot be used because our sensor does not supply the required data, and it was not our intent to provide a comprehensive implementation and comparison of all possible methods. We explain why we chose these models for comparison in the below sentence.

Lines 310-311: “We also implemented two alternative models that can use the signals from our PAA to extract the solar azimuth.”

8. Page 7 line 393 : authors compared errors to a compass with 36 polarimeters which has only been simulated and not tested experimentally. This kind of comparison is not very fair and participates to blur the real performances of the device.

Ans: We apologise for the confusion, which is probably due to the wrong wording in our manuscript. All of the results presented in this manuscript are measured, not simulated. Our interpolation method is the same as the rotating polariser technique used in many polarisation compass studies (i.e., Lambrinos et al., 2000; Dupeyroux et al., 2019; Smith, 2008; among others). As we rotate our robot around its main axis, each PAA performs an azimuthal scan. Each measured value is stored along with an azimuth value (measured using the IMU), which allows us to create a dataset of photodiode pulses associated to the different azimuths. These real recordings from the sky were then used to estimate the performance of the sensor when using different number of PAAs. The interpolation technique can be seen as moving average filter. We have now corrected the terminology in our text and replaced the word 'simulated' with 'reconstructed'.

Lines 173-180: *"Using this pooled dataset we were able to reconstruct the performance of the same basic design using different numbers of PAAs distributed evenly around the sensor ring (Supplementary Fig. S1d). This was achieved by determining the preferred direction of the reconstructed PAAs, then taking the median of the five responses from the dataset which were most closely clustered around that preferred direction (Supplementary Fig. S1c)."*

9. For all these reasons, I do not recommend this paper for a publication in Communications Engineering journal. The method is interesting but I do not understand why authors finally did not use the four directions available for each polarimetric device and thus apply the classical Stokes formalism that could certainly lead to better performances.

Ans: Using the Stokes formalism (or an equivalent) to extract the AoP and DoP would not be an original contribution. Many recent methods use the extracted AoP and DoP and focus on creating more accurate compass models, e.g. by adding IMU data, or applying machine learning to improve the estimate from a particular measurement device. Here we present a completely different design and model that doesn't need the calculation of the Stokes parameters nor the AoP and DoP. Instead, it uses current knowledge of the brain and visual processing of insects to simplify the algorithmic problem, moving the complexity to the design of the sensor. The miniaturising of such a sensor could result in less energy hungry and smaller celestial compass solutions. We hope that our responses to the reviewer's comments will convince them that our work is worth publishing.

Reviewer #3:

The paper by Gkaniyas et al. describes the design and testing of a robot that uses sensors inspired by insect photoreceptors to detect the polarization pattern and intensity gradients in the sky for spatial orientation. The robot shows superb sun compass performance even under unfavourable conditions such as different clouding or under canopies and thus impressively illustrates the power of insect brain mechanisms in sky compass navigation. The design has several advantages over previous technical solutions and is therefore an important contribution to the field of autonomously navigating vehicles. I have several comments that might help to further improve the manuscript.

1. You use the term “polarimeter” for your polarization axis analysers. According to Wikipedia, a polarimeter is a device to measure the angle of rotation caused by passing linearly polarized light through an optically active substance, which is clearly not how the term is used here. You might either introduce a new definition for the term or use an alternative term such as “polarization axis analyser” or “polarization angle analyser”.

Ans: The reviewer is right. The term ‘polarimeter’ has a specific meaning that is different from the one used here, thanks for spotting this. We followed the reviewer’s advice in using the term ‘polarisation axis analyser’ (PAA) instead throughout our manuscript.

2. Throughout the manuscript it seems to me that you use the terms “solar azimuth”, “solar position” and “solar meridian” somewhat interchangeably, but each of these terms describes different things. The solar position is defined by a horizontal component, termed solar azimuth and a vertical component, the solar elevation. The solar meridian, on the other hand, is the great circle in the sky passing through the sun and the zenith. It might be helpful to add a figure describing these terms. What your robot is apparently capable of doing is to navigate according to solar azimuth, but in the summary you say that it “allows accurate extraction of the sun’s position” (which would include its elevation). Is that so? If so, how precise is it in this respect?

Ans: We understand the confusion regarding the multiple terms. Although sometimes we really mean the position of the sun (azimuth and elevation), most of the times we use the terms ‘position’ or ‘location of the sun’ and ‘solar meridian’ to describe the solar azimuth. We have now changed all these terms to ‘solar azimuth’ to avoid this confusion, we use the ‘position of the sun’ only when we mean its elevation and azimuth, and use the word location only to describe locations on Earth. In the summary, we meant to say that we extracted the solar azimuth, and this is now corrected. Given these changes, we do not think an additional figure describing the differences between the solar meridian, azimuth and position is necessary.

3. Line 22: “anti-solar meridian”. This is not exactly correct. The degree of polarization is strongest along a circle in the sky that is 90° away from the sun.

Ans: It is true that the “anti-solar meridian” is a generalisation, which we used for simplicity. Indeed, the DoP is stronger on a circle in the sky at 90° away from the sun, but the strongest DoP on this circle is at the anti-solar median. The anti-solar meridian in the polarisation compass literature is usually described as a line that starts from the zenith (celestial pole), terminates at the nadir, and passes through the solar azimuth + 180°. We acknowledge that

using this term might be confusing for someone with a different background and therefore we have now updated our description to avoid any confusion.

Lines 19-23: *“Under clear skies, skylight intensity peaks at the visual position of the sun (solar azimuth and elevation), while the DoP is strongest in the opposite direction (at 90° from the sun and across the zenith).”*

Lines 70-74: *“Smith [40] observed that rotating a (simulated) POL-OP unit at an angle from the zenith breaks the 180° ambiguity and its response is strongest in the opposite direction of the solar azimuth (corresponding with the DoP pattern in the sky).”*

4. Lines 42-44: These are very general statements that may be true for many insects but clearly not for all. Extreme cases are e.g. cave insects which are blind. Likewise, dorsal rim areas have not been reported in all insect orders (see Labhart and Meyer 1999; *Microscopy Research and Technique* 47:368). So perhaps modify the statements by saying “...many insects..”

Ans: The comment of the reviewer is accurate. Our phrase “The eyes and brains of insects...” was referring to insects that actually have eyes and they use them to derive a compass. We didn’t use the word ‘many’ on purpose, as it would suggest that there are insects that have eyes and use them to derive a compass, but they don’t have the same organisation, which is not accurate. However, we understand that this might be interpreted in different ways, and we follow the reviewer's suggestion and changed this.

Lines 42-43: *“The eyes and brains of many insects evolved to provide an alternative solution.”*

5. Fig. 1 legend: Many images here seem to be taken from other sources, such as Fig. 1b,c,f and g. Please acknowledge these sources in the figure legend.

Ans: Although these figures were created by the authors for this manuscript, indeed, Fig. 1c, f and g took inspiration from figures in Gkaniyas et al. (2019). To the extent of our knowledge, Fig. 1b does not look similar to any of the figures in the literature. If the reviewer has found this in any other source, we would really appreciate it if they could share this information with us. We added the below line at the end of the caption to acknowledge the similarities.

“c, f, and g were adapted and modified from [28].”

6. Line 139: ...that points towards the solar meridian...” Is this true? The solar meridian is a great circle passing through the sun and the zenith. Do you rather mean “position of the sun” or “solar azimuth”?

Ans: The reviewer is correct about this and we changed this phrase accordingly.

Lines 137-138: *“The vectors were then averaged, yielding a mean vector that points towards the solar azimuth.”*

7. Figure 8: there is no reference in the figure legend acknowledging the source of this image or the origin of data in this image.

We thank the reviewer for noticing this. The fly brain in the background was borrowed from Hardcastle et al. (2021). The data shown here has come from reviewing Hardcastle et al. (2021) and Kind et al. (2021), as indicated in the text. These are now acknowledged in the figure caption.

“Adapted and modified from [46]; data from [46,47].”

8. Line 422: why not use the simpler term “medulla-tubercle (MeTu) neurons.”

Ans: This is now changed as the reviewer suggested.

9. Line 433: likewise, “tubercle-bulb (TuBu) neurons” would be a bit simpler.

Ans: This is now changed as the reviewer suggested.

10. Line 452 following: This paragraph is based on evidence from the fly *Drosophila*. Therefore, do not use the general statement on line 451 “Insects have been shown...” Specifically refer to *Drosophila*.

Ans: The reviewer is right that this has not been shown in all insects. We have now updated the text accordingly.

Lines 425-429: *“Fruit flies (*D. melanogaster*) have been shown to integrate their absolute compass with self-motion in two different stages of processing: first in the AOTu (optic flow input), and later in the EB (feedback from motor neurons and optical flow) [47,48].”*

11. Line 725, 727: The name of the author is “Lord Rayleigh” F.R.S. stands for “Fellow of the Royal Society” which is not part of the author’s name.

Ans: This is now changed as the reviewer suggested.

12. Line 731: The initials of this author are J.W. (not H.J. as indicated here).

Ans: This is now changed as the reviewer suggested.

REVIEWERS' COMMENTS:

Reviewer #1 (Remarks to the Author):

The revised version is well-written based on all the reviewers' concerns.

Reviewer #2 (Remarks to the Author):

The revision has been properly done by the authors to address the comments that I have raised. Therefore, I recommend this paper for publication in RA-L.

Only a minor comment : A scale must added in figures 1h and 9g to have an idea about the size.

Reviewer #3 (Remarks to the Author):

All of my comments were addressed to my full satisfaction. I have no further comments for improvement of this excellent paper.

Reviewer #1 (Remarks to the Author)

The revised version is well-written based on all the reviewers' concerns.

Reviewer #2 (Remarks to the Author)

The revision has been properly done by the authors to address the comments that I have raised. Therefore, I recommend this paper for publication in RA-L.

Only a minor comment : A scale must added in figures 1h and 9g to have an idea about the size.

Ans: Scales have now been added to the figures.

Reviewer #3 (Remarks to the Author)

All of my comments were addressed to my full satisfaction. I have no further comments for improvement of this excellent paper.